# Follicular Size Threshold for Ovulation Reassessed. Insights from Multiple Ovulating Dairy Cows

**DOI:** 10.3390/ani12091140

**Published:** 2022-04-28

**Authors:** Fernando López-Gatius, Mònica Llobera-Balcells, Roger J. Palacín-Chauri, Irina Garcia-Ispierto, Ronald H. F. Hunter

**Affiliations:** 1Transfer in Bovine Reproduction SLu, 22300 Barbastro, Spain; lopezgatiusf@gmail.com; 2Agrotecnio Centre, 25198 Lleida, Spain; 3Department of Animal Science, University of Lleida, 25198 Lleida, Spain; monicalloberabalcells@gmail.com (M.L.-B.); palacin11@hotmail.com (R.J.P.-C.); 4Sidney Sussex College, University of Cambridge, Cambridge CB2 3HU, UK; vhladfield@btinternet.com; 53 Pleasants Steading, Oxnam, Jedburgh TD8 6QZ, UK

**Keywords:** bovine, follicular size, ovulation failure, pregnancy rate, twins

## Abstract

**Simple Summary:**

The selection of a single ovarian follicle able to differentiate and ovulate is a phenomenon common to monovular species including humans. The selected follicle acquires the capacity to ovulate when it reaches a diameter of about 10 mm. In cows with a single follicle of ovulatory size, the probability of ovulation significantly increases with follicle diameter. However, two or more follicles of ovulatory size are often present at estrus. In cows with one follicle of ovulatory size and another follicle of 7–9 mm, the small follicle may, under certain circumstances, ovulate producing a pregnancy.

**Abstract:**

In *Bos. taurus* cattle, follicular deviation to dominance begins when the selected ovulatory follicle reaches a mean diameter of 8.5 mm. The dominant follicle acquires the capacity to ovulate when it reaches a diameter of about 10 mm. In this study, data derived from 148 cows in estrus with one follicle of ovulatory size and another of 7–9 mm, reveal that the small follicle has the capacity to ovulate alone or with the dominant follicle; thus, giving rise to a single or twin pregnancy. This indicates that a follicle of deviation size may ovulate in the presence of a follicle of ovulatory size.

## 1. Introduction

Cattle breeds are generally considered monovular. There are, nevertheless, circumstances in which there is an increased incidence of double ovulation and thus of twin pregnancy. In dairy cattle, it is widely accepted that twining compromises both the health of cows [1,2,3] and herd economy [4,5,6]. The economic impacts of twin pregnancies are likely to rise as multiple ovulation rates, particularly in older cows, have increased substantially over the past 30 years along with increased milk productivity [7,8,9]. Genetic progress and improvements in nutrition and management practices have led to a continuous increase in milk yield [10,11,12] suggesting that the multiple ovulation rate will continue to rise in parallel with milk production. These are cogent reasons to examine the factors that cause multiple pregnancies, as the mechanisms underlying multiple ovulations remain to be elucidated [12].

During the estrous cycle of cattle, two or three waves of follicular growth take place, of which only the last wave is ovulatory. Each wave leads to the development of a large dominant follicle and smaller subordinate follicles [13,14,15]. Follicular deviation to dominance begins when the largest follicle reaches a mean diameter of 8.5 mm [15]. The deviation mechanism consists of the reduction or cessation of growth of the subordinate follicles while the largest follicle increases in size. The process finishes either with ovulation or atresia of the dominant follicle [13,14,15]. During the ovulatory wave, the dominant follicle acquires the capacity to ovulate when it reaches a diameter of about 10 mm [16]. In cows with a single follicle of ovulatory size, the probability of ovulation significantly increases with follicle diameter [17,18]. However, two or more follicles of ovulatory size are often present at the time of artificial insemination (AI) and multiple corpora lutea (CLs) may be recorded in over 50% of older cows [19]. 

Multiple ovulations involve the presence of at least two ovulatory follicles in either the same ovary or in both ovaries. Cows with two or more ovulatory follicles at estrus may show different ovulation patterns to those with a single ovulatory follicle. In a recent study examining 316 cows with two follicles of ovulatory size at AI, individual follicle diameter could not be related to the likelihood of ovulation [18]. In the latter study, only the smaller follicle ovulated in 40.5% of cows [18]. This event of ovulation of only the smaller follicles could be related to two processes: first, a second deviation may occur in cows with co-dominant follicles [20]; and second, during a follicular wave there could be a switch in diameter rank between the dominant and largest subordinate follicle just before or at deviation [21]. However, another possibility could be associated with the ovulation of follicles smaller than 10 mm in multiple ovulating cows, this being the minimum diameter, linked to a capacity to ovulate in monovular cows [16]. The observation of unexpected additional CLs seven days after AI in cows with a single follicle of ovulatory size [18,22] reinforces this idea. The objective of the present study was to assess the possibility of double ovulation in cows with a single follicle of ovulatory size (10 mm or more) and a follicle smaller than 10 mm at estrus. 

## 2. Materials and Methods

### 2.1. Cows and Herd Management

The study population was derived from a Holstein dairy herd in northeastern Spain (latitude 41.13 N, longitude −2.4 E) in which repeat breeder cows were synchronized for fixed-time AI (FTAI) 5–11 days after the last confirmed estrus. Cows were considered repeat breeders when they did not become pregnant after 3 AI in the absence of detectable anatomic abnormalities [23,24]. Pedometers were used to confirm estrus (AfiFarm System; Afikim, Israel) in both FTAI and spontaneous estrus animals. Cows ≥ 60 days in milk showing estrus during the previous 3 weeks and with no CL were recorded as anestrous cows and received the same FTAI protocol at the same time as repeat breeders. The remaining cows were inseminated either at spontaneous estrus or following another FTAI protocol. During the study period (October 2020 to July 2021), the mean number of lactating cows and mean annual milk production was 145 and 12,450 kg per cow, respectively. The mean annual culling rate was 31%. Cows were grouped according to parity (primiparous versus pluriparous) and fed complete rations.

The herd was subjected to a weekly reproductive health program on the day of FTAI. The study population was selected among cows ready for service and inseminated in the weekly visit after ultrasound scanning: FTAI repeat breeders, FTAI anestrous cows, and cows inseminated at spontaneous estrus (first, second, or third AI). In this way, most of the cows should have a single ovulatory follicle at AI, both repeat breeders [25] and cows inseminated during early lactation [26]. Cows were included if they were healthy, confirmed by a body condition score of 2.5–3.5 on a scale of 1–5 [27], produced ≥ 30 kg of milk per day, and were free of clinical signs of diseases from insemination days −7 to +28. In all cows, the presence of a follicle of ovulatory size (10 mm or more) in the absence of luteal structures, a uterus that was highly turgid and contractile to the touch, and copious vaginal fluid, were used as reference to confirm estrus by ultrasound and palpation per rectum [28]. Cows were inseminated by two technicians using frozen-thawed semen from 8 bulls. If a cow returned to estrus before pregnancy diagnosis, its status was confirmed by examination per rectum, and the animal was inseminated at this time and recorded as non-pregnant. A cow was included only once in the study.

### 2.2. Fixed-Time AI Protocol and Ultrasound Exams

Repeat breeders and anestrous cows were synchronized for FTAI using a controlled internal drug-releasing device (CIDR, containing 1.38 g of progesterone (P4); Zoetis Spain SL, Spain). The CIDR was left in place for 5 days, and these animals were also given cloprostenol (500 µg i.m.; PGF Veyx Forte, Ecuphar, Spain) on CIDR removal. Then, 24 and 60 h later, the cows received a second cloprostenol and a GnRH dose (using the GnRH analogue dephereline: 100 µg gonadorelin acetate [6-D-Phe] i.m.; Gonavet Veyx, Ecuphar, Spain), respectively. Cows were inseminated 72 h after CIDR removal [29]. 

Ovarian follicular structures larger than 6 mm in diameter and the absence or presence of one or more CLs were assessed by ultrasound immediately before AI and 7 days later using a portable B-mode ultrasound scanner equipped with a 5–10 MHz transducer (E.I. Medical IBEX LITE; E.I. Medical Imaging, Loveland, CO, USA). Each ovary was scanned in several planes, moving the transducer along its surface to identify follicular and luteal structures; measurements and the number and location of both were recorded. Ovulation, confirmed by the presence of at least one mature CL, was assessed on Day 7 post-AI. A lack of high pixel intensity associated with a young CL [30,31] was used as reference to confirm the state of CL maturity. Follicular diameter was measured on the widest image of the follicle and calculated as the average of the greater and lower diameter measurements. The dimensions of a CL were recorded as the mean of two measurements approximating the greatest length and width. As the presence of a central cavity is not functionally important [32,33,34], cavity CL were measured just like solid CL. Scanning was also performed along the dorsal/lateral surface of each uterine horn for pregnancy diagnosis on Day 28 post-AI. Twin pregnancies were registered as unilateral (both embryos in the same uterine horn) or bilateral (one embryo in the right horn and its co-twin in the left horn).

### 2.3. Data Collection and Statistical Analyses

The pregnancy rate was defined as the percentage of cows that became pregnant out of the total number of cows in the corresponding group. The following data were recorded in each animal: parturition and AI dates; status (repeat breeder, anestrus or spontaneous estrus); lactation number (parity, primiparous vs. pluriparous); the number and location of follicles and CL (unilateral vs. bilateral in cows with two structures); days in milk at AI (DIM; <90 days postpartum versus ≥90 days postpartum); milk production at AI (mean production in the three days before AI) (low producers < 45 kg vs. high producers ≥ 45 kg); sire; AI technician; follicular size at AI; ovulation failure (absence of at least a mature CL seven days after AI); CL size seven days after AI; double ovulation (presence of two CLs seven days after AI); pregnancy 28 days post-AI for ovulating cows; and presence of twins in pregnant cows. The threshold for milk production was set as the median value of production recorded in primiparous cows. Three follicular groups were established: cows with a single follicle, cows with two bilateral follicles (one follicle in each ovary), and cows with unilateral follicles (two follicles in the same ovary). AI dates were used to assess the effects of season on subsequent reproductive performance. In our geographical region, there are only two clearly differentiated seasons: warm (May to September) and cool (October to April) [35,36]. Temperatures for the study period were from October 2020 to April 2021: 28 days with minimum temperature < 0 °C and 3 days with maximum temperature >25 °C (15.25 ± 2.3); from May to July 2021: 0 days with minimum temperature < 0 °C and 75 days with maximum temperature > 25 °C (29.8 ± 5.2).

The software package PASW Statistics for Windows Version 18.0 (SPSS Inc., Chicago, IL, USA) was used for data processing. Significance was set at *p* < 0.05. Variables are expressed as the mean ± standard deviation (S.D.). Overall reproductive performance in the groups was compared using the chi-square test (percentages) or ANOVA and Tukey post-hoc tests (means ± SD). Possible correlation between follicular size and CL size was examined in single-ovulating and bilateral double-ovulating cows. 

The effects of the presence of a small follicle on ovulation and pregnancy rates were analyzed by binary logistic regression. Two regression analyses were performed using ovulation and conception after AI as the dependent variables. The factors entered in the models as independent variables were repeat breeding, anestrus, follicular group (three classes: single, and bilateral or unilateral in cows with a small follicle), season of AI (warm), parity (pluriparous), days in milk (>90 days) and milk production at AI (high producers). For the dependent variable conception rate, only ovulating cows were included. Possible interactions between the presence of small follicles and the variables milk production, season, and parity were also examined. Regression analyses were conducted according to the method of Hosmer and Lemeshow [37].

Two further binary regression models were built using the presence of a small follicle as the dependent variable for all cows, and ovulation of the small follicle for cows with this structure. The factors entered in the models were those considered for the dependent variable ovulation described above except follicular group. In the case of the dependent variable ovulation of the small follicle, the site with respect to the follicle of ovulatory size (bilateral vs. unilateral) and diameter difference between follicles were added as factors. 

## 3. Results

Twelve cows with no follicles of ovulatory size (10 mm or more), 164 cows with two or more follicles of ovulatory size, and 139 cows with luteal structures were removed from the study. The final study population was comprised of 434 cows with a single follicle of ovulatory size along with the presence or not of a single follicle between 7 and 9 mm: 246 repeat breeders, 61 anestrous cows, and 127 cows showing spontaneous estrus. Of the 434 cows, 286 (65.9%) had a single follicle, 99 (22.8%) had two bilateral follicles (one follicle in each ovary) and 49 (11.3%) had two unilateral follicles (two follicles in the right or left ovary). All cows were submitted to AI. Mean milk production and days in milk at the time of AI, as well as the number of lactations and AIs, were 50 ± 9 kg, 133 ± 82 days, 3 ± 1.3 lactations, and 4 ± 2.1 AIs, respectively (mean ± SD). The independent variables recorded for each follicular group and their effects on each dependent variable are shown in Table 1.

Follicular diameter was significantly (*p* < 0.0001) greater for follicles of pre-ovulatory size than small follicles. As high positive correlation (r = 0.86; *p* < 0.0001) was recorded between follicle diameter and CL size in single-ovulating plus bilateral double-ovulating cows, CL derived from small follicles were identified in cows with two unilateral follicles. Of the total of 148 small follicles, 51 (34.5%) ovulated: 33 in the bilateral and 18 in the unilateral group. Ovulation occurred only in the small follicle in 17 cows: 12 in cows with bilateral and 5 in cows with unilateral follicles. CL size was significantly (*p* < 0.0001) greater for CLs derived from follicles of ovulatory size than those derived from small follicles. According to chi-square tests, cows with two follicles showed a significantly (*p* < 0.001) greater proportion of double ovulations (34/142, 16.9%) referred to the total of ovulating cows, and of twin pregnancies (5/44, 11.4%) referred to the total of pregnant cows, compared with cows with one follicle, in which both parameters were 0%.

### 3.1. Ovulation and Conception Rates

Seventeen cows (3.9%) failed to ovulate. These cows were subjected to a further FTAI protocol at ovulation diagnosis and their data were excluded from the subsequent analyses but maintained for the analyses of ovulation failure rate and presence of a small follicle. Based on binary logistic regression procedures, no factors were noted to affect ovulation. No young CLs were detected in ovulating cows. Of the 417 ovulating cows, 128 (30.7%) became pregnant. Table 2 provides the pregnancy rate, odds ratio, and 95% confidence interval for the total population of ovulating cows. The final model included only repeat breeding. Impacts of follicular group, anestrus, parity, days in milk, season and milk production were not significant, so these factors were not included in the final model. Using non-repeat breeders as the reference, repeat breeding resulted in a pregnancy rate significantly reduced (*p* < 0.0001) by a factor of 0.4. Nine conceptuses were derived from small follicles: 6 (3 as co-twins) from the bilateral and 3 (2 as co-twins) from the unilateral group. One cow, in the unilateral group, became pregnant from its small follicle (9 mm), underwent ovulation failure of the follicle of ovulatory size (18 mm) and only a CL of 12 mm was observed 7 days post-AI.

### 3.2. Presence and Ovulation of Small Follicles

Of 434 cows with a follicle of ovulatory size, 148 (34.1%) had a small follicle. Table 3 provides the incidence of the presence of a small follicle, odds ratio, and 95% confidence interval for the total population of cows. The final model included only repeat breeding. The impacts of anestrus, parity, days in milk, season, and milk production were not significant and these factors were not included in the final model. Taking non-repeat breeders as the reference, repeat breeding resulted in an incidence of the presence of a small follicle that was significantly (*p* = 0.001) reduced by a factor of 0.6. No factors were associated with the ovulation of a small follicle.

## 4. Discussion

In this study, we examined the effects of the presence of a small follicle (7–9 mm) in lactating dairy cows with a follicle of ovulatory size (10 mm or more) at estrus. To our knowledge, the presence and ovulation of a small follicle and possible associated factors following either natural or synchronized estrus have not been previously analyzed. A total of 34.5% (51/148) of the small follicles ovulated, with an associated pregnancy rate of 17.6% (9/51). Under our working conditions, there was no influence of synchronization, anestrus, repeat breeding, site with respect to the follicle of ovulatory size (bilateral vs. unilateral), and diameter difference between follicles on ovulation of the small follicle. Although we were unable to find a significant correlation between the variables analyzed and the ovulation of such follicles, we did observe a strong link between the presence of a small follicle and repeat-breeder syndrome.

Follicles with 7–9 mm sizes accompanying a dominant follicle of ovulatory size at estrus are often considered as atretic. Follicular selection to dominance depends on a transient increase in plasma concentrations of follicle-stimulating hormone (FSH) [38,39,40]. The selected follicle continues development in the setting of a profound drop in FSH level and may escape atresia, which is the fate of all other follicles that lack the capacity to make use of the low FSH concentrations [41,42,43]. As a dominant follicle matures, there is a transfer of dependency from FSH to LH supporting the establishment of dominance [13,14,44]. 

The selection of a single ovarian follicle for differentiation and ovulation is a phenomenon shared by monovular species including humans [39,45,46]. However, the presence of two or more co-dominant follicles (follicles of ovulatory size) in high producers can exceed 50% at the time of estrus [18]. In these cases, the largest follicle is not necessarily the one most likely to ovulate. Ovulation only of the smaller follicle may occur in over 40% of ovulating cows [18], 10–30% being an accepted figure among cows, mares, and women [46,47]. The rate of switching of dominance to the small follicle seems similar in the present study. Hence, ovulation only of the small follicle is observed in 12% (17/142) of ovulating cows. However, our results also suggest that follicles of deviation diameter (7–9 mm) behave differently to the smaller follicle in cows with two co-dominant follicles of ovulatory size. In prior work, we were able to relate double ovulation along with ovulation of the smaller follicle to the least size difference between the larger and smaller follicle in a study population of 316 cows with only two co-dominant follicles (cows with three or more follicles of ovulatory size were removed) [18]. This pattern could not be demonstrated here as the diameter difference between follicles was not found to influence ovulation of the small follicles. Studies are needed to elucidate mechanisms related to the ovulation of follicles of deviation size.

It is well documented that a positive correlation exists between follicular diameter at estrus and the size of its associated CL [48,49,50,51]. The high positive correlation in cows with bilateral follicles allowed us to identify the ovulation of small follicles in the unilateral group. Irrespective of this, double ovulation was recorded in 23.9% (34/142) of ovulating cows with two follicles, and twin pregnancies in 14.7% (5/34) of cows experiencing double ovulation. This double ovulation rate is within reported ranges of 12–30% [7,8,26,29,52,53]. We can, therefore, view the presence of deviation-size follicles at estrus as an additional factor favoring twin pregnancies. In fact, an increased proportion of small ovulatory follicles (8–10 mm) has been associated with multiple ovulations (from 2 to 6 ovulations) in beef cow breeds selected on the grounds of five generations of multiple pregnancies [49]. Although a high dose of LH was found by some authors to result in 0% ovulation of follicles of near deviation diameter (7–9 mm) in dairy cows [16], we would expect the high presence of small ovulatory follicles besides two or more ovulatory-size follicles at estrus in dairy cattle. In effect, an increased level of follicular recruitment and growth has been described for cows [54] and women [55] delivering twins.

As we might expect, the presence of small follicles was reduced in repeat breeders. Abnormal follicular dynamics is a common reproductive pattern related to repeat-breeder syndrome [56,57,58] and a low double ovulation rate has been associated with this disorder [25,26]. Repeat breeding emerged here as the only factor influencing the conception rate, and 56.7% (246/434) of cows in our study population were classified as repeat breeders. 

## 5. Conclusions

In the presence of a dominant follicle of ovulatory size (10 mm or more) at estrus, follicles measuring between 7 and 9 mm can ovulate alone or along with the dominant follicle, leading to single or twin pregnancies. In dairy cattle, the lower follicular size limit for ovulation needs revising.

## Figures and Tables

**Table 1 animals-12-01140-t001:** Independent variables recorded at the time of AI and effects of the three ovulatory follicle states on each dependent variable (*n* = 434).

Follicular Status ^(a)^	One Follicle (*n* = 286)	Two Bilateral Follicles (*n* = 99)	Two Unilateral Follicles (*n* = 49)
Independent variables ^(b)^			
Parity (pluriparous)	155 (54.2%)	53 (53.5%)	28 (57.1%)
Milk production (≥45 kg)	140 (49%)	60 (60.6%)	24 (49%)
Days in milk (≥90 days)	223 (78%)	68 (68.7%)	36 (73.5%)
Season (warm period: May–September)	91 (31.8%)	33 (33.3%)	12 (24.5%)
Repeat breeding	174 (60.8%)	48 (48.5%)	24 (49%)
Spontaneous estrus	82 (28.7%)	28 (28.3%)	17 (34.7%)
Diameter (mean ± SD) of follicles of ovulatory size (≥10 mm)	19.8 ± 6.1	19.7 ± 6.2 *	19 ± 6 *
Diameter (mean ± SD) of small follicles (7–9 mm)		7.9 ± 0.5 **	7.8 ± 0.8 **
Size (mean ± SD) of CL derived from follicles of ovulatory size	23.7 ± 7.5	23.7 ± 7.3 *	22 ± 8.2 *
Size (mean ± SD) of CL derived from small follicles		10.8 ± 0.8 **	11.6 ± 1.1 **
Dependent variables ^(c)^			
Ovulation failure	11/286 (3.8%)	4/99 (4%)	2/49 (4.1%)
Double ovulation ^(d)^	0/275 (0%) ***	21/95 (22.1%) ****	13/47 (27.7%) ****
Conception rate ^(d)^	84/275 (30.5%)	33/95 (34.7%)	11/47 (23.4%)
Twin pregnancy ^(e)^	0/84 (0%) ***	3/33 (9.1%) ****	2/11 (18.2%) ****

^(a)^ One follicle: cows with a single follicle of ovulatory size (10 mm or more); two bilateral follicles: cows with a single follicle of ovulatory size in an ovary and a follicle between 7 and 9 mm in the contralateral ovary; two unilateral follicles: cows with a single follicle of ovulatory size in an ovary and a follicle between 7 and 9 mm in the same ovary. ^(b)^ Values with different superscripts within columns denote significant differences detected by ANOVA and Tukey post-hoc tests (*, **: *p* < 0.0001). ^(c)^ Values with different superscripts within rows denote significant differences detected by the chi-square test or Fisher’s exact test (***, ****: *p* < 0.001). ^(d)^ In ovulating cows. ^(e)^ In pregnant cows.

**Table 2 animals-12-01140-t002:** Odds ratios of the pregnancy rate variables included in the final logistic regression model (*n* = 417).

Factor	Class	*n*	% Pregnancy	Odds Ratio	95% Confidence Interval	*p*
Repeat breeding (>3 AI)	No	72/179	40.2	Reference		
	Yes	56/238	23.5	0.4	0.26–0.6	<0.0001

R^2^ Nagelkerke = 0.14.

**Table 3 animals-12-01140-t003:** Odds ratios for the small follicle presence variables included in the final logistic regression model (*n* = 434).

Factor	Class	*n*	% Pregnancy	Odds Ratio	95% Confidence Interval	*p*
Repeat breeding (>3 AI)	No	76/188	40.4	Reference		
	Yes	72/246	29.3	0.6	0.41–0.91	0.001

R^2^ Nagelkerke = 0.15.

## Data Availability

The data presented in this study are available on request. These data are not publicly available to preserve the data privacy of the commercial farm.

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
