# Peer review of "Follicular Size Threshold for Ovulation Reassessed. Insights from Multiple Ovulating Dairy Cows"

_animals, 2022, doi:10.3390/ani12091140_

Round 1

Reviewer 1 Report

Review of the manuscript Frontiers 909446 entitled “Follicular Size Threshold for Ovulation Reassessed. Insights from Multiple Ovulating Dairy Cows

General comments

This manuscript addresses whether a follicle within deviation size could ovulate in the presence of a follicle of ovulatory size. In general, the manuscript is clear and well written. However, authors need to clarify how they tracked the fate of small ovulating follicles, when they were ipsilateral to a large follicle and resulted in a single CL and single pregnancy. A few minor suggestions are listed below.

Specific comments:

Line 20: In B. taurus cattle, follicular deviation....

Lines 74-75: Estrus detection was performed in cows undergoing FTAI, or only for those inseminated at spontaneous estrus?

Line 144: …two clearly differentiated seasons: …

Line 170: “…and 139 cows with luteal structures…” At the moment of AI?

Lines 174-175 and elsewhere: In this case, please consider using the words “ipsilateral” and “contralateral”, instead of “unilateral” and “bilateral”. Otherwise, it may sound like the same follicle is in both sides.

Line 176: All cows were submitted to AI.

Table 1 and line 200: When the number of occurrences within a class is below 5 (i.e., 0/84, 3/33, 2/11), the Fisher’s exact test shall be used, instead of the Chi-squared.

Line 206: Confusing. Did the authors mean “…in non-ovulation cows.”?

Lines 213 and 236: How did the authors inferred that the non-co-twin pregnancy from the ipsilateral group was from the small, rather than from the large follicle?

Lines 236-238: Under our working conditions, there was no influence of synchronization, reproductive condition (anestrus, repeat breeding), site with respect to the follicle of ovulatory size (ipsi vs contralateral), […] on ovulation of a small follicle.

Line 2412: Follicles with 7 to 9 mm…

Lines 257-258: Once again, if ovulations were not monitored, how do the authors know which follicle actually ovulated, in cows with both the larger and the smaller follicles at the same ovary? Or does this data refer to only cows with contralateral follicles?

Lines 258-260: Do the authors refer to the current or past results?

Author Response

Thank you very much for your comments

authors need to clarify how they tracked the fate of small ovulating follicles, when they were ipsilateral to a large follicle and resulted in a single CL and single pregnancy.

In material and methods we specify that: 286 (65.9%) had a single follicle, 99 (22.8%) had two bilateral follicles (one follicle in each ovary) and 49 (11.3%) had two unilateral follicles (two follicles in the right or left ovary). Cows with more than 2 follicles did not entered in the study. 

 A few minor suggestions are listed below.

Specific comments:

Line 20: In B. taurus cattle, follicular deviation.... Change done

Lines 74-75: Estrus detection was performed in cows undergoing FTAI, or only for those inseminated at spontaneous estrus? Sentence has been modified

Line 144: …two clearly differentiated seasons: … Done

Line 170: “…and 139 cows with luteal structures…” At the moment of AI? Yes. In some farms is normal to found a small CL at AI. This is why we decided to removed them from the study

Lines 174-175 and elsewhere: In this case, please consider using the words “ipsilateral” and “contralateral”, instead of “unilateral” and “bilateral”. Otherwise, it may sound like the same follicle is in both sides.

I think there is a confusion here. We mean in the same ovary (unilateral) and in both ovaries (bilateral). It has nothing to do with pregnancy, as it’s explained in material and methods. If we talk about pregnancy, we meant ‘in the same horn’. If we write ‘contralateral’ the reader may think ‘contralateral to CL’

Line 176: All cows were submitted to AI. Done

Table 1 and line 200: When the number of occurrences within a class is below 5 (i.e., 0/84, 3/33, 2/11), the Fisher’s exact test shall be used, instead of the Chi-squared. Done

Line 206: Confusing. Did the authors mean “…in non-ovulation cows.”? We changed the sentence

Lines 213 and 236: How did the authors inferred that the non-co-twin pregnancy from the ipsilateral group was from the small, rather than from the large follicle? We only use cows with two follicles. Then it’s simple to know that if, for example, a cow has two right ipsilateral follicles and it’s pregnant of two ipsilateral co-twins, the oocytes come from the small and the largest follicle.

Lines 236-238: Under our working conditions, there was no influence of synchronization, reproductive condition (anestrus, repeat breeding), site with respect to the follicle of ovulatory size (ipsi vs contralateral), […] on ovulation of a small follicle. Done

Line 2412: Follicles with 7 to 9 mm… Done

Lines 257-258: Once again, if ovulations were not monitored, how do the authors know which follicle actually ovulated, in cows with both the larger and the smaller follicles at the same ovary? Or does this data refer to only cows with contralateral follicles? We explain that question above

Lines 258-260: Do the authors refer to the current or past results? We changed the sentence

Reviewer 2 Report

The paper under review deals with the assessment the possibility of double ovulation in cows with a single follicle of ovulatory size (10 mm or more) and a follicle smaller than 10 mm at estrus. The topic of the investigation are in the scope of the journal. The evaluated paper contributes new information to the field. The study showed that a follicle of deviation size (7-9 mm) may ovulate in the presence of a follicle of ovulatory size (about 10 mm) and may give a twin pregnancy. The Authors concluded that the lower follicular size limit
for ovulation needs revising.

My main concern relates to the heterogeneous population of cows studied, including repeat breeders, anoestrus cows and cows showing spontaneous estrus. Repeat breeders and anestrous cows were synchronized for FTAI using a CIDR and cloprostenol.  Thus, the cows had different progesterone levels and follicles status, which could affect follicle development and the occurrence of double ovulation. Were the cows examined before synchronization started? Please consider this in the discussion. The authors state that there was no influence on ovulation of a small follicle of synchronization (line 236), but synchronization was not included as an independent variable in their models.

Author Response

Thank you very much for your comments

My main concern relates to the heterogeneous population of cows studied, including repeat breeders, anoestrus cows and cows showing spontaneous estrus. Repeat breeders and anestrous cows were synchronized for FTAI using a CIDR and cloprostenol.  Thus, the cows had different progesterone levels and follicles status, which could affect follicle development and the occurrence of double ovulation. Were the cows examined before synchronization started? Please consider this in the discussion. The authors state that there was no influence on ovulation of a small follicle of synchronization (line 236), but synchronization was not included as an independent variable in their models.

We analyzed synchronization protocol vs natural estrus as an independent variable, as it’s observed in Table 1. No differences were found.

Reviewer 3 Report

This is an interesting and well written manuscript; the information is well presented, the language is used correctly and very easy to understand.

Line 83: In the introduction, it is mentiones that the cows were grouped according to age, but instead of how old the cows were, they mentioned the parity: primiparous vs pluriparous. It´s recomended to switch the term parity instead of age.

In the discusión, the results and the interpretetion of them were mentioned, but there is not comparation between the studies done before, so it is suggested to make a contrast between what this study found and what other autors have described.

Author Response

Thank you for your corrections.

  • Line 83: In the introduction, it is mentioned that the cows were grouped according to age, but instead of how old the cows were, they mentioned the parity: primiparous vs pluriparous. It´s recommended to switch the term parity instead of age.

You are right, done

  • In the discussion, …., so it is suggested to make a contrast between what this study found and what other authors have described.

We are sorry, but there are no other authors studying the same topic in Holstein cows. That’s why we compare with other breeds and beef cows.

Reviewer 4 Report

The manuscript “Follicular size threshold for ovulation reassessed. Insights from multiple ovulating dairy cows” is well structured and interesting, its materials and methods are adequate, and the results obtained are of interest to the scientific community. Nonetheless, some corrections need to be done before publishing:

Grammar and writing style

L18: review the use of commas

L57: Add “of” before “cows”

L63-64: Revie the use of commas

L88: This sentence leads the reader to think that the mentioned parameters are what ensure the presence of a single ovulatory follicle, and no ultrasound was performed, but later it is stated that animals were subjected to ultrasound exams, maybe, it would be advisable to change this sentence.

L143-144: This could be improved by adding the mean weather of each season, to better define “warm” and “cool”

L214: Change “becoming” for “became”, also review the use of commas in this sentence

L236-238: Review the writing of this sentence, it is not clear

L281: Review if “in effect” should be “in fact”

The first paragraph of the results might be better represented in some graphic, where the information about the percentage of cows with each classification of follicles is presented. Also, the study population could be better presented as a graphic.

Finally, most of the references are older than five years, it is advisable to update the references were possible.

Author Response

Thank you for your comments

Grammar and writing style

L18: review the use of commas. Done

L57: Add “of” before “cows”. Done

L63-64: Revie the use of commas. Done

L88: This sentence leads the reader to think that the mentioned parameters are what ensure the presence of a single ovulatory follicle, and no ultrasound was performed, but later it is stated that animals were subjected to ultrasound exams, maybe, it would be advisable to change this sentence. Sentence has been changed

L143-144: This could be improved by adding the mean weather of each season, to better define “warm” and “cool”. We added a new sentence

L214: Change “becoming” for “became”, also review the use of commas in this sentence. Done

L236-238: Review the writing of this sentence, it is not clear. We changed the sentence

L281: Review if “in effect” should be “in fact”. We think here ‘in effect’ is more appropriate. Done.

The first paragraph of the results might be better represented in some graphic, where the information about the percentage of cows with each classification of follicles is presented. Also, the study population could be better presented as a graphic. 

We are sorry, we think that Table 1 is clearer than a graphic and includes practically the same information with statistics.

Finally, most of the references are older than five years, it is advisable to update the references were possible.

We agree. However, most studies on follicular dynamics as we can observe by ultrasound in reproductive controls are older than five years. Anyway, 22% 13/60 of the references used are recent (2016–2021)
